# Blocking SHP2 benefits FGFR2 inhibitor and overcomes its resistance in *FGFR2*-amplified gastric cancer

Yue Zhang[1], Hanbing Wang[1], Yutao Wei[2], Yunfeng Pan[1], Xueru Song[1], Jie Shao[1], Lixia Yu[1], Tao Shi[1]*, Yue Wang[1]*

[1]Department of Oncology, Nanjing Drum Tower Hospital, Affiliated Hospital of Medical School, Nanjing University, Nanjing, China; [2]Nanjing Drum Tower Hospital Clinical College of Traditional Chinese and Western Medicine, Nanjing University of Chinese Medicine, Nanjing, China

## eLife Assessment

Based on the perceived low efficacy of current therapies targeted to FGFR2 in gastric cancer (GC), the authors investigate an approach that combines SHP2 inhibition with existing FGFR2 inhibitors. The data were largely collected and analysed using **solid** and validated methodology. There is some **useful** data regarding combination therapy in a new clinical cohort, which supports previous studies that have reported the potential of targeting RTKs together with phosphatases.

**\*For correspondence:**
taoshi@smail.nju.edu.cn (TS);
wangyue2012nju@163.com (YW)

**Competing interest:** The authors declare that no competing interests exist.

**Abstract** Fibroblast growth factor receptor 2 (FGFR2) is an important member of receptor tyrosine kinase (RTK) family. *FGFR2* amplification occurs at a high frequency in gastric cancer (GC) and has been proven to be closely associated with poor prognosis and insensitivity to chemotherapy or immunotherapy. Current FGFR2-targeted therapies have limited efficacy. Hence, how to enhance efficacy and reverse resistance are urgent problems clinically. Src homology region 2-containing protein tyrosine phosphatase 2 (SHP2) serves as the shared downstream mediator of all RTKs and a prominent immunosuppressive molecule. In this study, we identified *FGFR2* amplification in 6.2% (10/161) of GC patients in our center. Then we showed that dual blocking SHP2 and FGFR2 enhanced the effects of FGFR2 inhibitor (FGFR2i) in *FGFR2*-amplified GC both in vitro (human GC cell lines) and in vivo (mouse xenograft tumor models) via suppressing RAS/ERK and PI3K/AKT pathways. We further showed that it overcame FGFR2i resistance by reversing the feedback activation mediated by other RTKs and continuously suppressing FGFR2-initiated downstream pathways. Notably, SHP2 blockade could suppress PD-1 expression and promoted IFN-γ secretion of CD8[+] T cells, enhancing the cytotoxic functions of T cells in tumor immune microenvironment. Overall, our findings suggest that dual blocking SHP2 and FGFR2 is a compelling rationale with both targeted treatment and immune regulation for *FGFR2*-amplified GC.

## Introduction

Gastric cancer (GC) ranks as the third leading cause of cancer death globally (*Ajani et al., 2022*; *Smyth et al., 2020*; *Wei et al., 2020*). Due to the absence of specific early signs, over 70% of GC patients are diagnosed at an advanced stage (*Guan et al., 2023*; *Song et al., 2017*). Concurrently, treatments for advanced GC predominantly include chemotherapy, targeted therapy, radiotherapy, and immunotherapy. Anti-programmed cell death protein 1 (PD-1) antibody nivolumab plus chemotherapy has been approved by the Food and Drug Administration (FDA) as the standard first-line treatment for

advanced GC (*Janjigian et al., 2021*; *Smyth et al., 2020*; *Zhao et al., 2022*). Meanwhile, the Cancer Genome Atlas (TCGA) has classified GC patients into four subtypes, including Epstein–Barr virus (EBV) type, microsatellite instability (MSI) type, genomic stability (GS) type, and chromosomal instability (CIN) type (*The Cancer Genome Atlas Research Network, 2014*). Among them, EBV and MSI types are more likely to benefit from immunotherapies, while CIN and GS types tend to be less responsive. Fortunately, the unique composition structure of CIN type provides potential anti-tumor targets. CIN type is in the majority, accounting for about 50%, with distinct aneuploidy and focal amplification of receptor tyrosine kinases (RTKs) as its molecular feature (*The Cancer Genome Atlas Research Network, 2014*). RTKs are common tumor-driven genes, including human epidermal growth factor receptor 2 (*HER-2*), fibroblast growth factor receptor 2 (*FGFR2*), epidermal growth factor receptor (*EGFR*) and so on *Du and Lovly, 2018*; *Paul and Hristova, 2019*. The combination of anti-HER-2 antibody trastuzumab and chemotherapy has been approved by the FDA as the first-line treatment for HER-2-positive advanced GC (*Kang, 2023*; *Smyth et al., 2020*; *Yang et al., 2020*). In the meantime, other RTKs, especially FGFR2, have also displayed huge potential as anti-tumor targets.

FGFR2 is also a representative member of the RTK family. Several studies have documented that FGFR2 blockade can inhibit the occurrence and development of malignant tumors by regulating PI3K/AKT, RAS/ERK and JAK/STAT pathways (*Babina and Turner, 2017*; *Gordon et al., 2022*; *Katoh and Nakagama, 2014*). It is worth noting that *FGFR2* amplification has been identified in up to 7.7% of advanced GC patients (*Jogo et al., 2021*), and is closely associated with poor prognosis and limited response to chemotherapy and immunotherapy (*Ahn et al., 2016*; *Hur et al., 2020*; *Kim et al., 2019b*; *Koh et al., 2019*). Currently, there have been several approaches for FGFR2 inhibition in *FGFR2*-amplified GC. For instance, Bemarituzumab (FPA144) is a specific anti-FGFR2b monoclonal antibody (*Katoh et al., 2024*). However, in a phase II clinical trial (NCT03694522), it has been demonstrated that Bemarituzumab monotherapy did not statistically significantly improve progression-free survival (PFS) in FGFR2b-selected GC patients (*Wainberg et al., 2022*). Besides, small molecule inhibitors targeting FGFR2 are also primary methods for *FGFR2*-amplified GC treatment. AZD4547 is an FGFR-selective inhibitor targeting FGFR1-3, which has been widely used as an FGFR2 inhibitor in researches focusing on *FGFR2*-amplified tumors (*Jang et al., 2017*; *Xie et al., 2013*). In a previous phase II clinical trial (NCT01795768), AZD4547 achieved an overall response rate (ORR) of 33% in patients with previously treated advanced *FGFR2*-amplified gastroesophageal cancer, and a median PFS of responding patients of 6.6 months (*Pearson et al., 2016*). Based on its distinguished therapeutic effects, AZD4547 was granted by the FDA as an orphan drug for GC. However, subgroup analysis indicated that robust single-agent response to AZD4547 was only observed in high-level *FGFR2*-amplified cancers in this trial (*Pearson et al., 2016*). And in another phase II clinical trial (NCT01457846), AZD4547 failed to significantly improve patients' PFS compared to paclitaxel (3.5 months vs. 1.8 months, p=0.9581) as a second-line treatment for advanced gastric adenocarcinoma (*Van Cutsem et al., 2017*). Given that inhibiting FGFR2 alone is often difficult to achieve ideal therapeutic effects, identifying suitable combinations is crucial to the treatment of *FGFR2*-amplified GC.

Although RTK targeted therapies have been proven to have certain anti-tumor therapeutic effects, drug resistance of RTK inhibitors remain a common problem. One of the main reasons for RTK inhibitors resistance is the activation of downstream signaling pathways mediated by bypass RTKs. Previous research has revealed that the downstream pathway feedback upregulation caused by EGFR activation led to FGFR2 inhibitor resistance in intrahepatic cholangiocarcinoma (iCCA) with *FGFR2* fusion (*Wu et al., 2022*). Src homology region 2-containing protein tyrosine phosphatase 2 (SHP2) is the common downstream factor of RTKs including FGFR2, acting as a central node between FGFR2 and PI3K/AKT, RAS/ERK, or JAK/STAT pathways (*Chen et al., 2016*; *Sodir et al., 2023*). As the shared molecule downstream of all RTKs, SHP2 inhibitor and its combination with RTK inhibitors were reported to be ideal strategies for treating a large class of RTK-driven cancers, including *EGFR*-amplified, *KRAS*-amplified, *KRAS G12C*-mutated cancers and so on (*Chen et al., 2016*; *Fedele et al., 2021*; *Wong et al., 2018*). Therefore, we speculate that the combination of FGFR2 inhibitor and SHP2 inhibitor is likely to alleviate FGFR2 inhibitor resistance by inhibiting the feedback activation of bypass RTK pathways.

Meanwhile, SHP2 is also a significant downstream molecule of PD-1 (*Chen et al., 2016*; *Song et al., 2022*). SHP2 has been proven to suppress T cell activation by inactivating TCR and CD28 signaling (*Li et al., 2015*), which are two costimulatory signals mediating T cell development and

maturation (*Ai et al., 2020*; *Liu et al., 2020*). Numerous preclinical studies have revealed that SHP2 inhibition can inhibit tumors by enhancing anti-tumor immunity (*Liu et al., 2017*; *Zhao et al., 2019*). Accordingly, we suppose that simultaneously blocking FGFR2 and SHP2 has the potential to suppress tumor growth through both targeted intervention and immune activation.

Overall, we aim to explore the combined anti-tumor capacity and potential mechanisms of co-inhibiting FGFR2 and SHP2 in *FGFR2*-amplified GC, providing a 'two-pronged' combination therapy mode with targeted and immune dual effects for GC patients with *FGFR2* amplification.

## Results

### Recurrent *FGFR2* gene amplification in Chinese GC patients

161 GC samples collected from Nanjing Drum Tower Hospital were performed with NGS. The top 5 most significantly altered genes were *TP53* (61%), *ARID1A* (25%), *CDH1* (16%), *ERBB2* (12%), and *PIK3CA* (12%) (*Figure 1—figure supplement 1*). Meanwhile, *CCNE1* (8%), *ERBB2* (7%), *FGFR2* (6.2%), *MET* (6%), and *CCND1* (6%) were the top 5 most frequently amplified genes in the cohort (*Figure 1A and C*). Notably, the proportion of *FGFR2* amplification ranked third after *CCNE1* and *ERBB2* among all amplified genomes (*Figure 1A*). We can find *FGFR2*-amplified GC patients also occupied quite a high portion in TCGA-STAD cohort (15/295; 5%) (*Figure 1B*), further emphasizing the prevalence of *FGFR2*-amplified GC patients.

Moreover, in our cohort, patients with *FGFR2* amplification tended to present later TNM stages, although not statistically significant in difference (*Supplementary file 1*; *Figure 1—figure supplement 2*). Also, patients harboring *FGFR2* amplification in TCGA exhibited higher FGFR2 mRNA expression levels (p<0.0001) (*Figure 1—figure supplement 3*). And, FGFR2 mRNA expression showed a positive correlation with PTPN11 mRNA expression (p=0.0496, $R^2$=0.01468) (*Figure 1—figure supplement 4*), the chief coding gene of SHP2, suggesting a potential co-overexpression of SHP2 in patients with *FGFR2* amplification.

### SHP099 enhances the anti-tumor effects and overcomes the resistance of FGFR2 inhibitors in *FGFR2*-amplified GC

*FGFR2*-amplified GC cell lines SNU-16 and KATOIII were detected to exhibit relatively higher FGFR2 expression levels compared to other human GC cell lines (*Figure 2—figure supplement 1*). Then, the anti-tumor capacity of FGFR2 inhibitor, SHP2 inhibitor monotherapy or combination therapy was detected in vitro. It demonstrated that the combination administration of AZD4547 and SHP099 significantly suppressed cell proliferation compared to AZD4547 monotherapy in both KATOIII (p<0.0001) and SNU-16 (p<0.0001) (*Figure 2A and B*). Moreover, in groups treated with high concentration of AZD4547, the combination therapy notably enhanced cancer cell apoptosis in both KATOIII (p=0.0001 Combo vs. SHP099; p<0.0001 Combo vs. AZD4547) and SNU-16 (p<0.0001 Combo vs. SHP099; p=0.0017 Combo vs. AZD4547) (*Figure 2C and D*). The above results confirm that the combination of SHP099 and AZD4547 has a synergistic effect on tumor cell killing and apoptosis in *FGFR2*-amplified GC in vitro.

To unravel the mechanisms underlying the anti-tumor effects of combination therapy, we investigated the expression levels of key proteins of PI3K/AKT and MAPK pathways in KATOIII following drug treatment for 1 hour or 48 hours. While the combination of 3 nM AZD4547 and 10 μM SHP099 might not have led to a stronger inhibition of phospho-FGFR and phospho-SHP2 compared to 10 nM AZD4547 single-agent treatment, it induced a more pronounced suppression of downstream signal pathways of FGFR2, particularly phospho-ERK1/2 (*Figure 2E*). Interestingly, levels of phospho-p38, which is a crucial cascade mediating another MAPK pathway (*Lee et al., 2020*), remained unaffected by the medication. It is worth noting that although phospho-FGFR and phospho-SHP2 were further suppressed over time, AZD4547 single-agent treatment failed to sustain downstream signal suppression after 48 hours. In contrast, the combination treatment could continuously inhibit the downstream signaling molecules and overcome the feedback activation caused by prolonged treatment duration. Similar regulations of these phospho-proteins were also observed in another *FGFR2*-amplified GC cell line SNU-16 (*Figure 2F*).

### The synergistic efficacy of combining AZD4547 with SHP099 in primary tumor cells derived from an FGFR2 inhibitor-resistant GC patient

A 49-year-old Asian woman was diagnosed with poorly differentiated gastric signet-ring cell carcinoma (SRCC) through gastroscopy in November 2020. Subsequently, she underwent laparoscopic

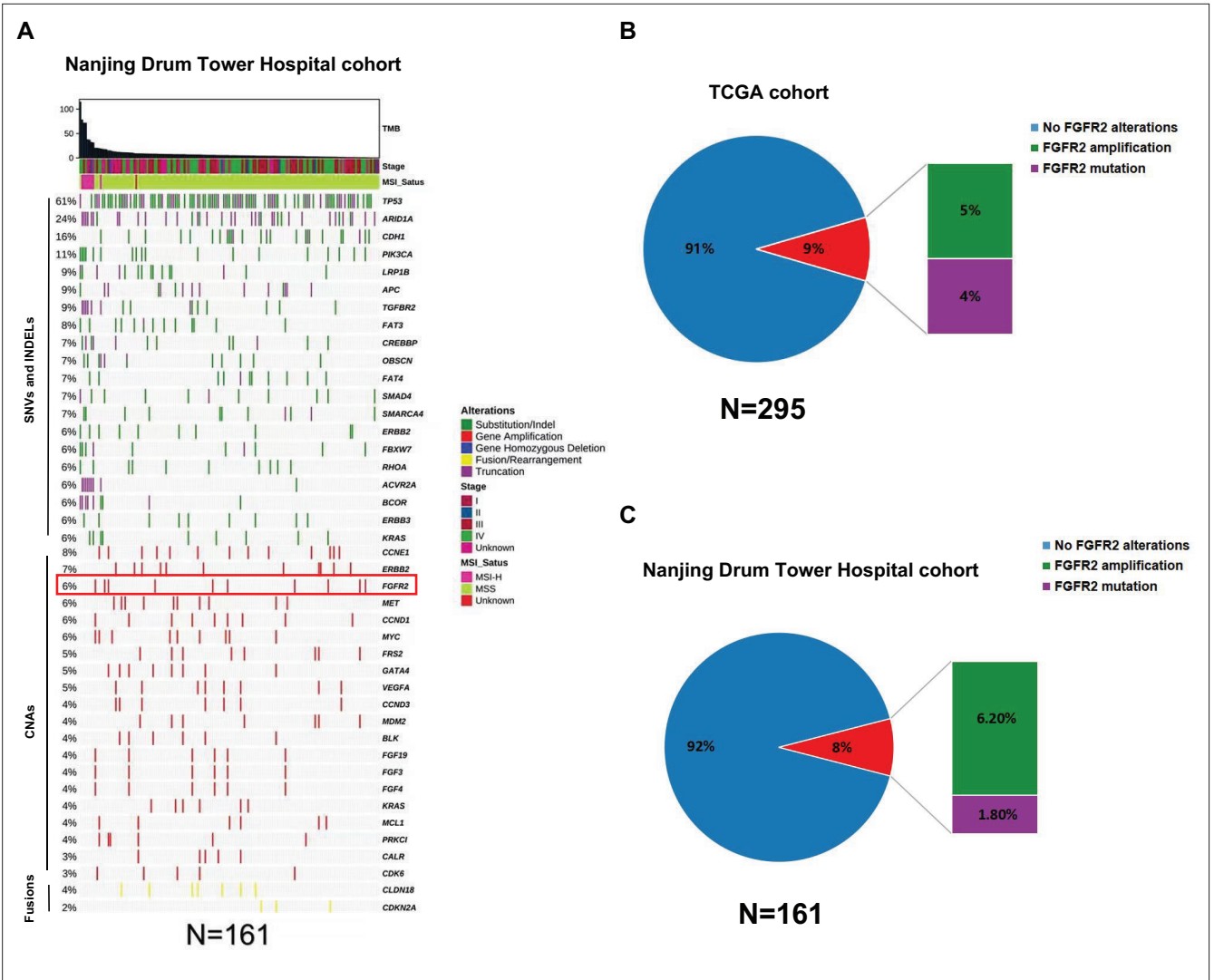

**Figure 1.** Recurrent *FGFR2* gene amplification in Chinese GC patients. (**A**) Overview of genomic alterations in GC patient samples collected in Nanjing Drum Tower Hospital (n=161). The patient samples are shown on the x-axis. Information on TMB, stage, MSI-status, and significantly altered genes is shown on the y-axis, with frequency of each alteration annotated on the left of the waterfall plot. Pie chart displaying the proportion of *FGFR2* alterations in (**B**) TCGA STAD cohort (n=295) and (**C**) Nanjing Drum Tower Hospital cohort (n=161).

The online version of this article includes the following figure supplement(s) for figure 1:

**Figure supplement 1.** Overview of genomic alterations in GC patient samples collected in Nanjing Drum Tower Hospital (n=161).

**Figure supplement 2.** Proportions of different AJCC stages and TNM stages among *FGFR2*-amplified (n=9) group and *FGFR2*-unamplified group (n=124) from Nanjing Drum Tower hospital cohort.

**Figure supplement 3.** FGFR2 mRNA expression levels were analyzed between *FGFR2*-amplified group (n=13) and *FGFR2*-unamplified group (n=250) from TCGA-STAD cohort (unpaired *t*-test).

**Figure supplement 4.** Correlation between PTPN11 and FGFR2 mRNA expressions among samples from TCGA-STAD cohort (n=263; linear regression *t*-test).

radical gastrectomy and received a 12-cycle biweekly chemotherapy regimen of Docetaxel and Tegafur from December 2020 to January 2021. However, serum alpha-fetoprotein (AFP) concentration began to rise in September 2021, and liver tumor metastasis rapidly ensued. Given that her tumor tissue NGS revealed an amplified *FGFR2* copy number of 87.1, she commenced FGFR2 inhibitor treatment in March 2022 (*Figure 3A*). During the medication period, she initially received partial response (PR) and experienced a transient reduction in her liver lesion, but rapidly came to progressive disease (PD) in December 2022, manifesting multiple lesions (*Figure 3B*). Similarly, serum AFP

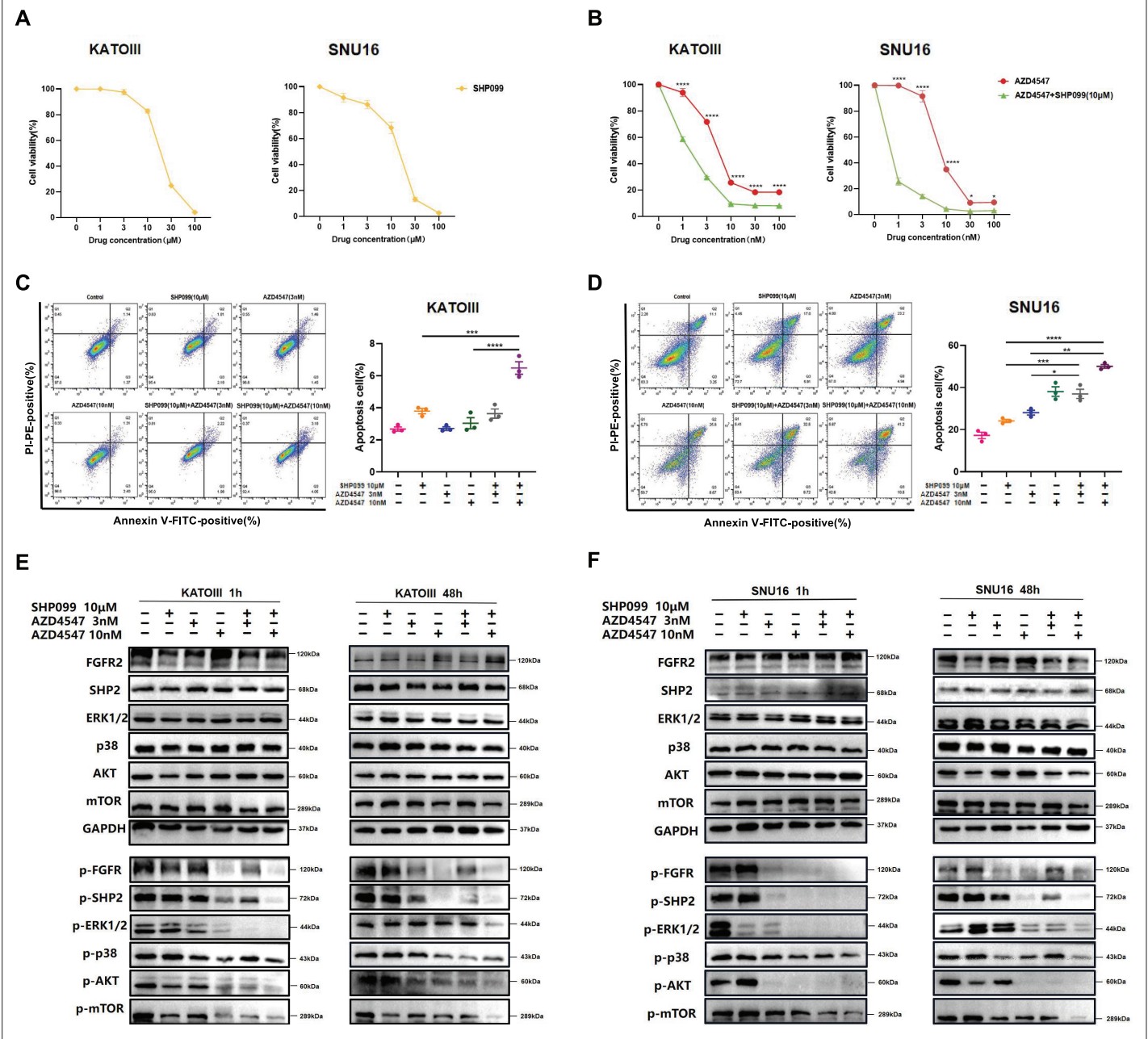

**Figure 2.** SHP099 enhances the anti-tumor effects and overcomes the resistance of FGFR2 inhibitors in *FGFR2*-amplified GC. Sensitivity of KATOIII and SNU-16 to (**A**) SHP099, (**B**) AZD4547, or combination therapy with different concentration gradients (n=4; two-way ANOVA). Effects of different treatments on cell apoptosis of (**C**) KATOIII and (**D**) SNU-16 after 48-hour drugs incubation (n=3; one-way ANOVA). (**E**) KATOIII and (**F**) SNU-16 were incubated with vehicle, SHP099 10 μM, AZD4547 3 nM or combination therapies for 1 hour or 48 hours, then cell lysates were immunoblotted for phospho-FGFR and total-FGFR2, phospho-SHP2 and total-SHP2, phospho-Erk and total-Erk, phospho-p38 and total-p38, phospho-AKT and total-AKT, and phospho-mTOR and total-mTOR. Data are shown as mean ± SEM. *p<0.05, **p<0.01, ***p<0.001, ****p<0.0001. p-Values are determined by ordinary one-way ANOVA or two-way ANOVA.

The online version of this article includes the following source data and figure supplement(s) for figure 2:

**Source data 1.** PDF file containing original western blots for *Figure 2E and F*, indicating the relevant bands.

**Source data 2.** Original file for western blots displayed in *Figure 2E and F*.

**Figure supplement 1.** Expression levels of total-FGFR2 in different human GC cell lines were detected by western blotting.

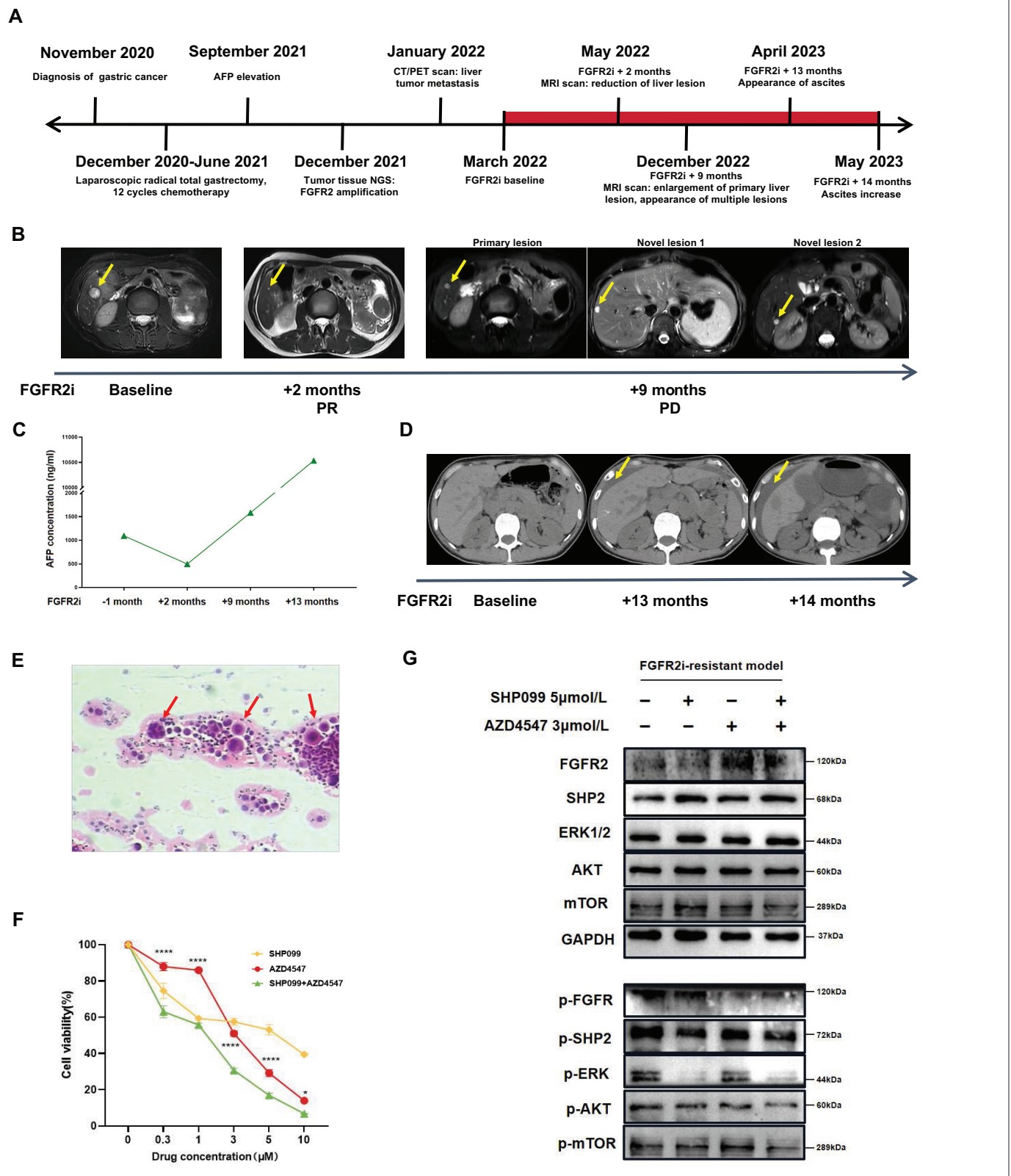

**Figure 3.** The synergistic efficacy of combining AZD4547 with SHP099 in primary tumor cells derived from a FGFR2 inhibitor-resistant GC patient.
(**A**) Schematic of therapeutic process of the *FGFR2*-amplified GC patient who was previously sensitive to FGFR2 inhibitors but quickly became resistant.
(**B**) Representative CT and MRI images of recurrent liver lesions at different time points. Liver lesions are indicated by yellow arrows. (**C**) Representative CT images displaying the occurrence and progression of ascites. Ascites are indicated by yellow arrows. (**D**) Pathology of malignant asites from this

*Figure 3 continued on next page*

Figure 3 continued

FGFR2 inhibitor-resistant patient. Tumor cells are indicated by red arrows. (**E**) AFP concentration variations of the patient during FGFR2 inhibitors medication. (**F**) Sensitivity of cancer cells from FGFR2 inhibitor-resistant patient's asites to FGFR2 inhibitor AZD4547, SHP2 inhibitor SHP099 or combined administration with different concentration gradients (n=4; two-way ANOVA). Data are shown as mean ± SEM. *p<0.05, ****p<0.0001. p-Values are determined by two-way ANOVA. (**G**) Tumor cells from asites were incubated with vehicle, SHP099 5 μM, AZD4547 3 μM or combination therapy for 1 hour, then cell lysates were immunoblotted for phospho-FGFR and total-FGFR2, phospho-SHP2 and total-SHP2, phospho-Erk and total-Erk, phospho-AKT and total-AKT, and phospho-mTOR and total-mTOR.

The online version of this article includes the following source data for figure 3:

**Source data 1.** PDF file containing original western blots for **Figure 3G**, indicating the relevant bands.

**Source data 2.** Original file for western blots displayed in **Figure 3G**.

level decreased after 2 months of medication but subsequently continued to rise, reaching up to 10,000 ng/ml (**Figure 3C**). As a result, we inferred that the patient initially exhibited sensitivity to FGFR2 inhibitors but swiftly developed resistance. After taking FGFR2 inhibitor for 13 months, the patient developed ascites (**Figure 3D**), and pathological examination confirmed the presence of tumor cells in her ascites (**Figure 3E**). To further evaluate the synergistic efficacy of dual blocking FGFR2 and SHP2 in patient-derived GC cells, we treated tumor cells derived from the patient's ascites with AZD4547, SHP099, or their combination in vitro for 48 hours. Consistently, the data indicated that FGFR2 inhibitor-resistant tumor cells from her ascites were more sensitive to the combination therapy compared to AZD4547 monotherapy (p<0.0001) (**Figure 3F**). To elucidate the underlying mechanisms, we detected the expression levels of key proteins related to FGFR2-initiated PI3K/AKT and RAS/ERK pathways after incubating tumor cells with different treatments for 1 hour. Interestingly, we did not observe a significant downregulation of phospho-SHP2 in groups with addition of SHP099. However, combination therapy exhibited relatively stronger inhibitory effects on phospho-FGFR and its downstream molecules, including phospho-AKT and phospho-mTOR. Furthermore, the addition of SHP099 significantly suppressed phospho-ERK1/2 (**Figure 3G**), suggesting that SHP099 may overcome FGFR2 inhibitor resistance mainly by suppressing RAS/ERK pathway. This case suggests that the combination of AZD4547 and SHP099 has potential application value in clinical FGFR2 inhibitor-resistant patients.

## The combination of SHP099 and AZD4547 has significant anti-tumor effects in SNU-16 xenograft nude mice

To evaluate the tumor-killing capacity of combining SHP099 with AZD4547 in *FGFR2*-amplified GC in vivo, we established a subcutaneous SNU-16 xenograft model in nude mice (**Figure 4A**). As anticipated, combination administration remarkably diminished tumor growth in vivo compared to both AZD4547 (p<0.0001, **Figure 4C**; p=0.0258, **Figure 4D**) and SHP099 (p=0.0005, **Figure 4C**; p=0.0832, **Figure 4D**) monotherapy (**Figure 4B–D**, **Figure 4—figure supplement 1**). Moreover, there were no notable signs of weight loss or drug toxicity observed (**Figure 4E**, **Figure 4—figure supplement 2**). Furthermore, consistent with in vitro studies, similar regulatory patterns of molecules downstream of FGFR2 were observed in protein samples derived from mouse tumor tissues. We found that combination therapy led to a more significant suppression than single-agent treatment, especially in phospho-AKT and phospho-mTOR (**Figure 4F**). To sum up, these results confirm the curative effects, elucidate the action mechanisms, and demonstrate the safety profile of combining SHP099 with AZD4547 in an in vivo model.

## SHP099 activates CD8[+] T cells and promotes their tumor-killing capacity in vitro

Our evidence suggested that the *FGFR2*-amplified group may be less likely to benefit from common immune therapies (**Fuchs et al., 2018**; **Kim et al., 2018**; **Shitara et al., 2018**; **Wang et al., 2021**). We observed that PD-L1 Combined Positive Score (CPS)-negative (CPS < 1) occurred more frequently in *FGFR2*-amplified GC patients compared to unamplified group in Nanjing Drum Tower Hospital cohort (**Figure 5A**; **Figure 5—figure supplement 1**). Similarly, in the TCGA-STAD cohort, *FGFR2*-amplified GC patients exhibited lower PD-L1 mRNA expression (**Figure 5B**). Meanwhile, there existed no *FGFR2*-amplified patients with high microsatellite instability (MSI-H) in both Nanjing Drum Tower

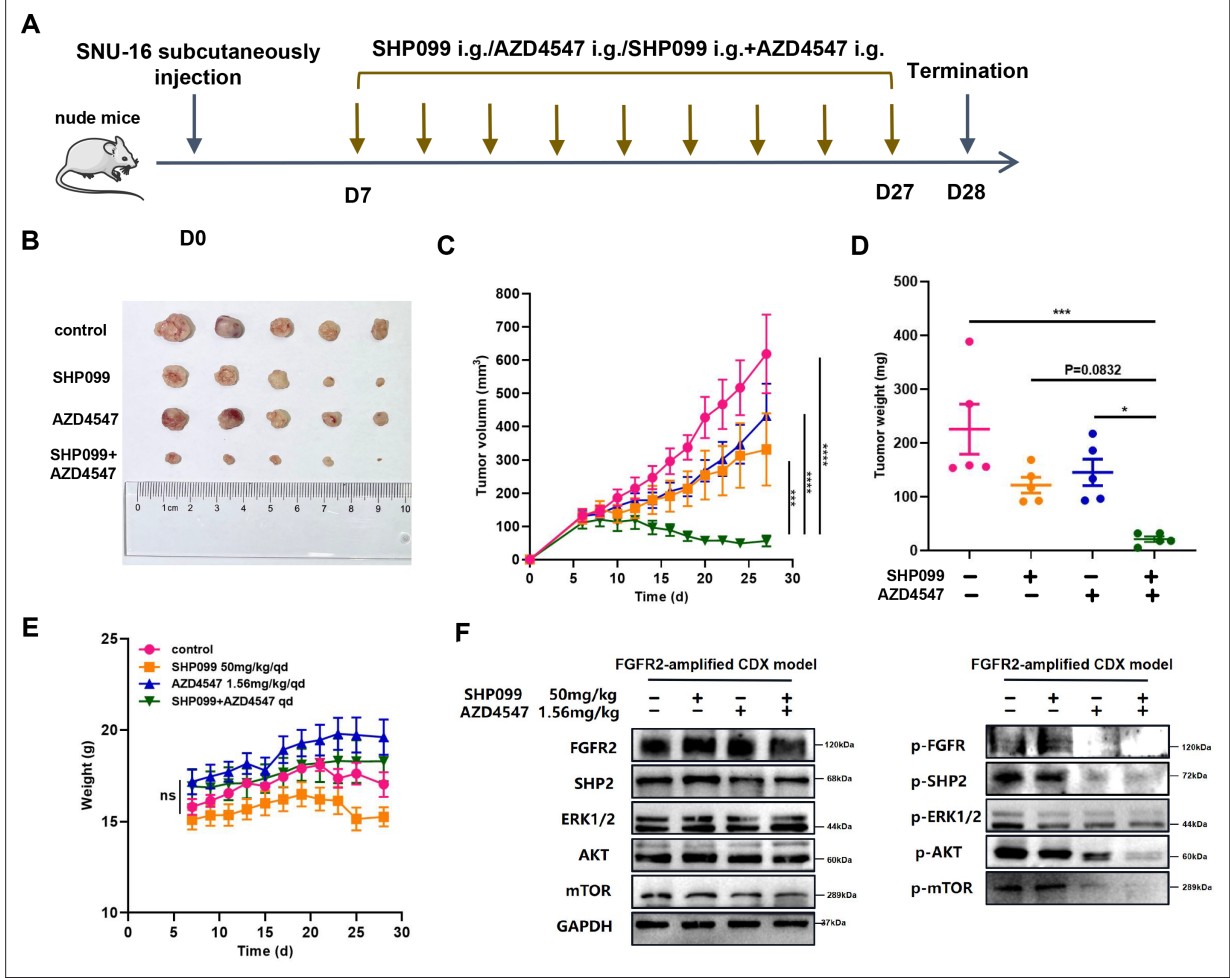

**Figure 4.** The combination of SHP099 and AZD4547 has significant anti-tumor effects in SNU-16 xenograft nude mice. BALB/c nude mice were injected with $1 \times 10^7$ SNU-16 cancer cells and received different formulations by oral gavage every day upon tumor volumes reached 100–150 mm³. (**A**) Schematic of SHP099 and AZD4547 therapeutic schedule in *FGFR2*-amplified SNU-16 subcutaneous xenograft model. (**B**) Representative images of tumors (n=5). (**C**) Tumor volumes (n=5; two-way ANOVA), (**D**) tumor weights (n=5; one-way ANOVA) and (**E**) body weights (n=5; two-way ANOVA) of different groups. (**F**) Tumors were harvested 6 hours after the last dose of drugs, and cell lysates from tumor tissues were immunoblotted for phospho-FGFR and total-FGFR2, phospho-SHP2 and total-SHP2, phospho-Erk and total-Erk, phospho-AKT and total-AKT, and phospho-mTOR and total-mTOR. Data are shown as mean ± SEM. ns, not significant, *p<0.05, **p<0.01, ***p<0.001, ****p<0.0001. p-Values are determined by ordinary one-way ANOVA or two-way ANOVA.

The online version of this article includes the following source data and figure supplement(s) for figure 4:

**Source data 1.** PDF file containing original western blots for *Figure 4F*, indicating the relevant bands.

**Source data 2.** Original file for western blots displayed in *Figure 4F*.

**Figure supplement 1.** Tumor volume of individual mice in control, SHP099, AZD4547, SHP099+AZD4547 group.

**Figure supplement 2.** SHP2 inhibition combined with FGFR2 inhibition is safe in SNU-16 xenograft nude mice model.

Hospital cohort (*Figure 5A*; *Figure 5—figure supplement 2*) and TCGA-STAD cohort (*Figure 5C*). Furthermore, in the TCGA-STAD cohort, *FGFR2*-amplified GC patients showed lower tumor mutation burden (TMB) levels versus unamplified group (*Figure 5—figure supplement 3*).

Considering the immunosuppressive role of SHP2 under PD-1/PD-L1 signaling, we also evaluated the effects of SHP099/AZD4547 combination therapy on T-cell immune activation in vitro. After incubating human PBMCs with different drug therapies for 24 hours in the presence of human anti-CD3 and anti-CD28, we observed that only groups with the addition of SHP099 could significantly induce the production of IFN-γ in CD8$^+$ T cells (p=0.0133 SHP099 vs. CD3/CD28 group; p=0.0167 Combo vs. CD3/CD28 group) (*Figure 5D*, *Figure 5—figure supplement 4*). Similarly, utilizing CBA detection,

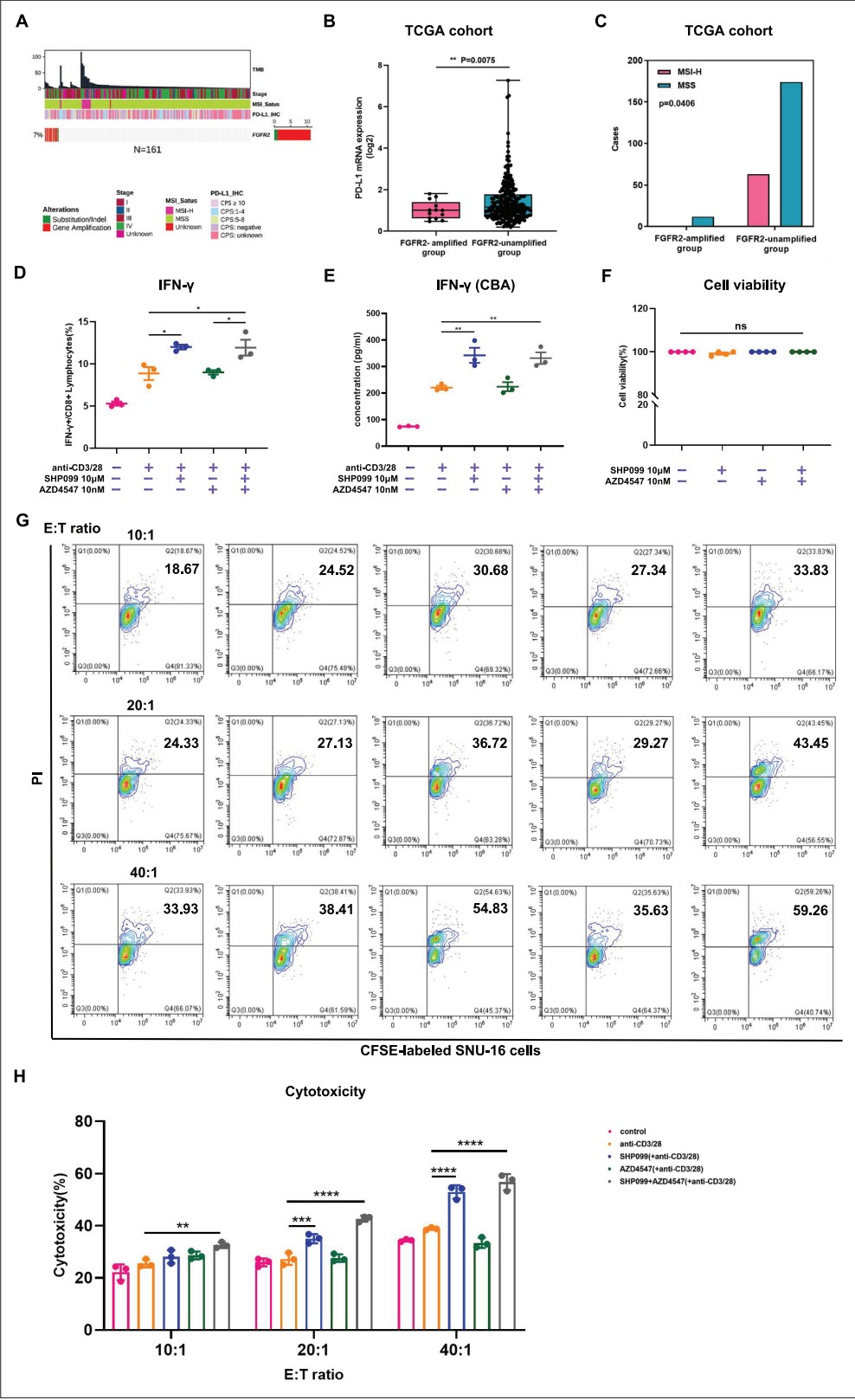

**Figure 5.** SHP099 activates CD8[+] T cells and promotes their tumor-killing capacity in vitro. (**A**) Overview of FGFR2 alterations individually in GC patients from Nanjing Drum Tower hospital cohort. (**B**) PD-L1 mRNA expression levels were analyzed between *FGFR2*-amplified group (n=13) and *FGFR2*-unamplified group (n=250) from TCGA-STAD cohort (Welch's *t*-test). (**C**) Proportions of MSI-H or MSS in *FGFR2*-amplified group (n=12) and *FGFR2*-unamplified

*Figure 5 continued on next page*

*Figure 5 continued*

group (n=237) from TCGA-STAD cohort (Fisher's exact test). Human peripheral blood mononuclear cells (PBMCs) were incubated with different administrations in the presence of 0.25 µg/ml human anti-CD3 and 1 µg/ml human anti-CD28. Proportions of (**D**) IFN-γ/CD8+ cells were detected by flow cytometry assay after 24 hours of drugs incubation (n=3; one-way ANOVA). (**E**) Expression levels of IFN-γ in cellular supernatant were measured by Cytometric Bead Array (CBA) assay after 24 hours of drugs incubation (n=3; one-way ANOVA). (**F**) Cell viability of human PBMCs after 48 hours of drugs incubation (n=4; one-way ANOVA). (**G, H**) After 48 hours of advance drugs stimulation in human PBMCs, cytotoxicities of human PBMCs against SNU-16 at different E:T ratios of 10:1, 20:1, 40:1 were analyzed by flow cytometry assay after CFSE/PI staining (n=3; two-way ANOVA). Data are shown as mean ± SEM. ns, not significant, *p<0.05, **p<0.01, ***p<0.001, ****p<0.0001. p-Values are determined by Welch's *t*-test, Fisher's exact test, ordinary one-way ANOVA or two-way ANOVA.

The online version of this article includes the following figure supplement(s) for figure 5:

**Figure supplement 1.** PD-L1 CPS outcomes among *FGFR2*-amplified group (n=6) and *FGFR2*-unamplified group (n=78) from Nanjing Drum Tower hospital cohort (Pearson's Chi-square test).

**Figure supplement 2.** MSI status among *FGFR2*-amplified group (n=10) and *FGFR2*-unamplified group (n=150) from Nanjing Drum Tower hospital cohort (Fisher's exact test).

**Figure supplement 3.** TMB levels were analyzed between *FGFR2*-amplified group (n=15) and *FGFR2*-unamplified group (n=274) from TCGA-STAD cohort (Wilcoxon test).

**Figure supplement 4.** Proportions of IFN-γ/CD8+ cells were detected by flow cytometry assay after 24 hours of drugs incubation.

**Figure supplement 5.** Proportions of IFN-γ/CD4+ cells were detected by flow cytometry assay.

---

we confirmed that PBMCs treated with SHP099 monotherapy (p=0.0053 vs. CD3/CD28 group) or combined therapy (p=0.01 vs. CD3/CD28 group) secreted more IFN-γ (*Figure 5E*). It is worth noting that a notable upregulation in CD4+ T cells was also observed when incubated with combination treatment (p<0.0001 vs. CD3/CD28 group) (*Figure 5—figure supplement 5*). Importantly, the cell viability of human PBMCs was not affected by drug treatment (*Figure 5F*). Besides, T cells pre-stimulated by SHP099 monotherapy (p<0.0001 vs. CD3/CD28 group) or combination therapy (p<0.0001 vs. CD3/CD28 group) exhibited a more potent tumor-killing ability when incubated with *FGFR2*-amplified SNU-16 cells in vitro (*Figure 5G and H*). Based on the above results, our data suggest that SHP099 may also synergize with FGFR2 inhibitor by eliciting CD8+ T-cell anti-tumor immunity and improving the inhibitory TIME.

## Discussion

*FGFR2* amplification is a common form of genetic variations in GC, and it was identified in 5% of GC patients in TCGA-STAD cohort. Meanwhile, in our study, *FGFR2* amplification occupied around 6.2% of GC patients in Nanjing Drum Tower Hospital cohort. And, it is conclusively shown that *FGFR2* amplification of GC is closely related to poor prognosis. Previous study demonstrated that *FGFR2* amplification was significantly associated with lymph node metastasis, poor tumor differentiation, and worse survival in GC (*Kim et al., 2019a*). Consistently, *FGFR2*-amplifed GC patients were observed to exhibit more advanced tumor stages in our cohort. It suggested that *FGFR2* amplification is a potential therapeutic target for GC.

Although FGFR2-targeted therapy has made some progress in GC treatment, there still exist problems such as low drug sensitivity and susceptibility to drug resistance. We reported a typical clinical case of an *FGFR2*-amplified gastric SRCC patient, showing symptoms of liver metastasis less than one year after radical gastrectomy. The patient received FGFR2 inhibitors and achieved PR after 2-month medication, with a duration of response only 7 months. This case also indicates that *FGFR2*-amplifed GC patients tend to have poorer prognosis and develop resistance to FGFR2 inhibitors in a relatively short period.

In terms of the mechanisms of FGFR2 inhibitors resistance as early as 2013, the research in *Cancer Discovery* revealed the potential causes of FGFR2 inhibitors resistance in *FGFR3*-mutated bladder cancer patients. It demonstrated that the bypass EGFR-mediated signaling activation would reduce the efficacy of FGFR inhibitors. As a result, the drug sensitivity of *FGFR3*-mutated bladder cancer patients to FGFR inhibitors was limited, leading to the occurrence of FGFR inhibitors resistance

(*Herrera-Abreu et al., 2013*). Another research conducted a more in-depth exploration of this mechanism. Researchers found that in iCCA with *FGFR2* fusion, feedback activation of downstream RAS/ERK, PI3K/AKT pathways mediated by EGFR alternative pathway is a significant contributor to FGFR2 inhibitors' resistance. By combining EGFR inhibitor with FGFR2 inhibitor, the feedback activation of downstream pathways was successfully relieved (*Wu et al., 2022*). The existing research has recognized the crucial role played by feedback activation in RTK inhibitors resistance. And, it cannot be ruled out that there are other bypass RTK activation besides EGFR, such as HER-2, cMET, FGFR3, etc (*Huang et al., 2024*; *Ryan et al., 2020*).

SHP2 is a common molecule downstream of all RTKs. Several studies have documented that SHP2 inhibitor displayed superior anti-tumor capacity and synergistic combination efficacy with RTK inhibitors in RTK-driven tumors (*Chen et al., 2016*; *Fedele et al., 2021*; *Wong et al., 2018*). Since SHP2 transduces signaling from all RTKs, we speculate that SHP2 inhibition can not only directly promote the anti-tumor effects of FGFR2 inhibitors, but also overcome FGFR2 inhibitors resistance by blocking all possible alternative RTK pathways. Our study first established that SHP099 can effectively promote the direct anti-tumor capacity of AZD4547 in different *FGFR2*-amplified GC cell lines by further suppressing downstream RAS/ERK and PI3K/AKT pathways. Meanwhile, SHP099 could continuously inhibit upstream and downstream signaling molecules and overcome FGFR2 inhibitor resistance in long-term drug-stimulated tumor cells. On the contrary, AZD4547 monotherapy exhibited obvious feedback activation of downstream pathways. We also validated the combined efficacy of SHP099 and AZD4547 in an SNU-16 cell-derived xenograft (CDX) nude mouse model, and no significant drug toxicity was observed. For further research, we isolated primary tumor cells from the FGFR2-inhibitor-resistant patient's ascites. It was observed that the combination treatment with SHP099 could enhance the anti-tumor efficacy of AZD4547 by inhibiting downstream RAS/ERK and PI3K/AKT pathways. By dual blocking FGFR2 and SHP2, we successfully overcame FGFR2 inhibitors resistance by suppressing FGFR2-iniated downstream pathways in FGFR2 inhibitor-resistant patient's cancer cells. These findings provide a potentially effective approach for overcoming FGFR2 inhibitors resistance.

Previous studies only focused on drug resistance caused by bypass signaling activation of a specific RTK gene, while our study utilized SHP2 inhibitor to counteract feedback activation of downstream signaling pathways caused by all RTK activation, providing a widely applicable combination therapy model. Apart from the targeted tumor-killing capacity, we also validated the immune regulation role of SHP2 inhibition in treating *FGFR2*-amplified GC patients. *FGFR2*-amplified patients in TCGA-STAD cohort tended to have poorer PD-L1 mRNA expression and lower TMB levels compared to the unamplified group, while MSS occurred more frequently in the *FGFR2*-amplified group. These characteristics are inextricably correlated with reduced responsiveness to anti-PD-1/PD-L1 therapy and poorer immune infiltration (*Fuchs et al., 2018*; *Kim et al., 2018*; *Shitara et al., 2018*; *Wang et al., 2021*). Meanwhile, SHP2 serves as a downstream molecule in the PD-1 signaling pathway. It has been confirmed that SHP2 can be recruited and bound to PD-1, thus inhibiting cytotoxic T cells activation by suppressing CD28 and TCR signals (*Ai et al., 2020*; *Liu et al., 2020*). A previous study demonstrated that T-cell SHP2-deficient mice bearing colitis-associated cancer showed enhanced cytotoxicity of CD8[+] T cells and reduced tumor sizes (*Liu et al., 2017*), corroborating that SHP2 inhibition can facilitate T cell anti-tumor immunity. In addition, the combination of SHP2 inhibitor and anti-PD-1 antibody has been proven to have synergistic anti-tumor effects in colon cancer by activating cytotoxic CD8[+] T cells and normalizing TIME (*Zhao et al., 2019*). The above evidence confirmed the enormous potential value of inhibiting SHP2 in regulating anti-tumor immunity. Although several researches have revealed that SHP2 inhibitor can be ideal synergistic combination partners for various RTK inhibitors from the perspective of targeted therapy, including MET (*Pudelko et al., 2020*), EGFR (*Liu et al., 2021*; *Sun et al., 2020*; *Xia et al., 2021*), and RET inhibitors (*Lu et al., 2024*), currently no in-depth study focusing on its immune regulatory function has been conducted. Our study demonstrated for the first time in vitro that SHP099 combined with AZD4547 downregulated the T-cell exhaustion molecule PD-1 and up-regulated IFN-γ secretion in CD8[+] T cells, leading to an enhanced tumor-killing capacity of cytotoxic T lymphocytes in *FGFR2*-amplifed GC cell line.

Taken together, our study collectively affirmed that the combination of SHP099 and AZD4547 can not only promote the targeted tumor-killing effects and overcome AZD4547 drug resistance, but also activate T cell immunity in *FGFR2*-amplified GC (*Figure 6*). We have validated the utility and feasibility of combining SHP2 inhibitor to FGFR2 inhibitor in *FGFR2*-amplified GC patients, providing

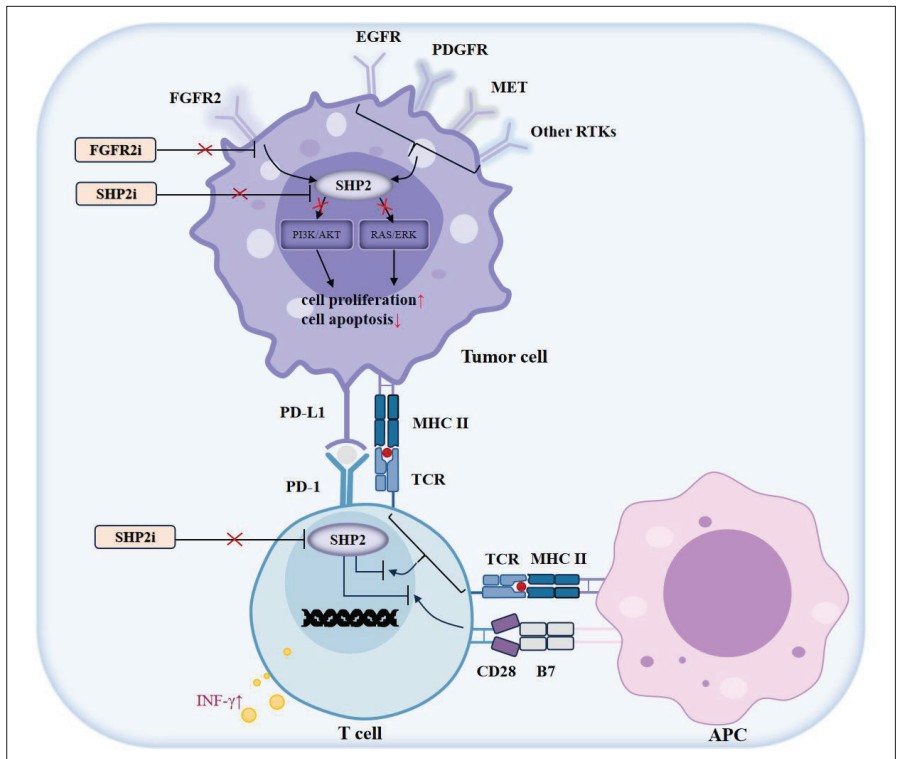

**Figure 6.** Schematic of the mechanisms of blocking SHP2 and FGFR2 in inhibiting tumor progression by targeted therapy and immune intervention. On one hand, SHP2 inhibitor can boost the tumor-cytotoxicity effects of FGFR2 inhibitor by inhibiting PI3K/AKT and RAS/ERK pathways in *FGFR2*-amplified GC, and overcome FGFR2 inhibitor resistance caused by feedback activation. On the other hand, SHP2 inhibitor can activate CD8[+] T cells to kill tumor cells, suggesting its synergizing effects with FGFR2 inhibitor by enhancing T cell-mediated anti-tumor immune responses. Our results demonstrate the utility and feasibility of combining SHP2 inhibitor to FGFR2 inhibitor in GC patients with *FGFR2* amplification.

an effective combination mode with both targeted intervention and immune regulation. Moreover, we also furnished sufficient evidence for the combination of RTK inhibitors and SHP2 inhibitors for all RTK-driven pan-cancer treatment.

## Materials and methods
### Patient samples and next-generation sequencing
We collected gastroscopy biopsy or surgically collected specimens of 161 GC patients from January 2016 to August 2023 in Nanjing Drum Tower Hospital. Next-generation sequencing (NGS) and analysis were performed in OrigiMed (Shanghai, China) based on 450 YuanSu gene panel. At least 50 ng DNA were extracted from formalin-fixed paraffin-embedded (FFPE) tumor tissues using QIAamp DNA FFPE Tissue Kit. Genes capture and sequencing were performed at a mean depth of 1000× by Illumina NextSeq 500 (San Diego, CA, USA). Gene alterations, including substitution, gene amplification, gene homozygous deletion, gene fusion and truncation were evaluated. TMB was estimated by calculating somatic non-synonymous mutations per megabase in each patient. The study was carried out in compliance with the code of ethics of the World Medical Association (Declaration of Helsinki) and approved by the Ethics Committee of Nanjing Drum Tower Hospital (No. 2021-324-01).

### Mice
Four- to six-week-old female BALB/c nude mice were purchased from GemPharmatech Co. Ltd. Mice were kept in specific pathogen-free animal facilities at the Comprehensive Cancer Centre of Nanjing Drum Tower Hospital.

## Cells

Human GC cell line KATOIII was donated by the Nanjing University Chao Yan laboratory. Human GC cell lines SNU-16 (ATCC CRL-5974), MKN45 (KANGBAI CBP60488), NUGC4 (KANGBAI CBP60493), HGC27 (KANGBAI CBP60480), and SNU601 (KANGBAI CBP60507) were purchased from the Cell Bank of the Chinese Academy of Sciences (Shanghai, China). All cells were cultured in Roswell Park Memorial Institute (RMPI) 1640 medium supplemented with 10% fetal calf serum (FBS) and 1% penicillin-streptomycin at 37°C with 5% $CO_2$. Primary GC cells were derived from ascites of an *FGFR2*-amplified GC patient who was resistant to FGFR2 inhibitors. Primary human peripheral blood mononuclear cells (PBMCs) were separated from human peripheral blood donated by two healthy people by Ficoll (TBD, LTS1077) and induced by 80 IU/ml IL-2 (PeproTech, 200-02). Human samples used in this study obtained patients' consent and the approval of the Ethics Committee of Nanjing Drum Tower Hospital (No. 2021-324-01). All cell lines have been proven to be free of mycoplasma contamination through STR analysis.

## CCK-8 assay

Cell Counting Kit-8 (CCK-8) (Vazyme, A311-01/02) assay was used to evaluate the effects of AZD4547 (Selleck, S2801), SHP099 (MedChemExpress, HY-100388) and combination administration on cancer cell proliferation. 3000 GC cells were seeded in 96-well plates with 100 µl of 10% FBS 1640 medium, and treated with SHP099, AZD4547 or combination therapy on the second day. After 4 days incubation, 10 µl CCK-8 was added to each well and incubated for 2 hours. Asorbance was measured at 450 nm with microplate reader. Cell viability=[(OD_Drug-OD_Blank)/(OD_Control-OD_Blank)]×100%.

## Cell apoptosis analysis

Annexin V-FITC/PI Apoptosis Detection Kit (Vazyme, A211-01) was used to detect drug effects on cancer cell apoptosis. GC cells were seeded in 12-well plates at a density of $1 \times 10^5$ cells per well and incubated with different formulations for 2 days. Afterward, cells were collected and suspended in 100 µl binding buffer containing 2.5 µl Annexin V-FITC and 2.5 µl propidium iodide (PI) for 10 minutes in the darkness. Samples were run by a flow cytometer and analyzed with FlowJo (RRID:SCR_008520).

## Western blotting

SNU-16 and KATOIII were seeded in 6-well plates at a density of $5\times10^5$ cells per well with 10% FBS 1640 medium and were incubated with SHP099, AZD4547, or combination therapy for 1 hour or 2 days. Primary human tumor cells derived from ascites of an *FGFR2*-amplified GC patient were incubated with different administrations for 1 hour. Mouse tumor tissues were collected after 6 hours of the last drug administration and homogenized after snap freezing. Then tumor tissues and cells were lysed in cell lysis buffer (NCM Biotech, WB3100) containing 1% protease and phosphatase inhibitors (NCM Biotech, P002). 10 µg of protein was used to detect the expression levels of FGFR2-initiated downstream signaling molecules by SDS-PAGE, electro-transfer, and immunoblotting with specific antibodies. The following antibodies were used: from Cell Signaling Technology, phospho-FGFR Tyr653/654 (Cell Signaling Technology Cat# 3476, 1:1000), SHP2 (3397T, 1:2000), phospho-SHP2 Tyr542 (3751T, 1:1000), ERK1/2 (4695T, 1:1000), phospho-ERK1/2 Thr202/Tyr204 (4370T, 1:2000), p38 (8690T, 1:1000), phospho-p38 Thr180/Tyr182 (4511T, 1:1000), AKT (9272S, 1:1000), phospho-AKT Ser473 (4060S, 1:2000), mTOR (2983T, 1:1000), phospho-mTOR Ser2448 (5536S, 1:1000), GAPDH (5174S, 1:1000), anti-mouse IgG, HRP-linked antibody (7076P2, 1:2000); from Santa Cruz Biotechnology, FGFR2 (sc-6930, 1:500); from Biosharp, Goat Anti-Rabbit IgG, HRP-linked Antibody (BL003A, 1:5000).

## Cell surface marker staining

PBMCs were incubated with different therapies in the presence of 0.25 µg/ml human anti-CD3 antibody (InVivoMAb, #BE0001-2) and 1 µg/ml human anti-CD28 antibody (InVivoMAb, #BE0248) for 24 hours, and then stained with FITC-anti-CD8 (BD Biosciences, 555634), Percp-Cy5.5-anti-CD4 (Biolegend, 317428) and PE-anti-PD-1 (Biolegend, 329906) for 30 min. Samples were run using a flow cytometer. The cellular supernatants were collected and detected by Cytometric Bead Array (CBA) human IFN-γ kit (BD Biosciences, 558456).

## Intracellular staining

The intracellular interferon-γ (IFN-γ) expression in CD8[+] T cells was detected by a BD Biosciences intracellular staining kit according to the instructions. After different processing for 24 hours, PMBCs were firstly stained with FITC-anti-CD8 and Percp-Cy5.5-anti-CD4 on the cell surface for 30 minutes, and then stained with PE-IFN-γ after fixation and permeabilization. Samples were then run using a flow cytometer and analyzed with FlowJo (RRID:SCR_008520).

## Cytotoxicity assay

To evaluate the tumor-killing capacity of drug-stimulated T cells in vitro, T cells were pretreated with different formulations for 2 days. SNU-16 cells labeled with carboxyfluorescein succinimidyl ester (Sigma, 21888) were seeded into ultralow 96-well plates at a density of $2 \times 10^4$ cells per well as the target cells. Drug-stimulated T cells were then added into the well at an E:T ratio of 10:1, 20:1, 40:1 as the effector cells and incubated with target cells for 9 hours. Afterward, cells were stained with propidium iodide (Sigma, 537059) for 10 minutes in the darkness and analyzed with flow cytometer.

## Xenograft tumor models

Four- to six-week-old female BALB/c nude mice (T cell deficient) were subcutaneously injected with $1 \times 10^7$ SNU-16 cells suspended in 100 μL PBS with 50% Matrigel (BD Biocoat, 356234). When the tumor volume reached approximately 100–150 mm$^3$, mice were randomized into four groups (n=5 per group) and started on treatment with PBS, SHP099 50 mg/kg, AZD4547 1.56 mg/kg, SHP099 50 mg/kg plus AZD4547 1.56 mg/kg. The dosage of medications used in animal experiments referred to previous research (*Wong et al., 2018*; *Xie et al., 2013*). Drugs were delivered by oral gavage every day for 21 days. Tumor dimensions were measured every other day and tumor size was calculated as 0.5 × length × width$^2$. All mice were killed on day 28 after tumor implantation. Tumor tissues were harvested for further analysis. Given the experimental design, randomization and blinding were deemed unfeasible and thus omitted. Nevertheless, tumor analysis was conducted in a blinded fashion. All animal experiments were approved by the Institutional Animal Care and Use Committee of Drum Tower Hospital (approval number: 2022AE01029).

## Statistical analysis

Large-scale analysis was conducted using data from the public TCGA database (https://pubmed.ncbi.nlm.nih.gov/25079317/), with further data processing and exploration performed via the cBioPortal platform. GraphPad Prism (RRID:SCR_002798) was used to conduct statistical analysis and construct graphics. Data are presented as the means ± standard error of the mean (SEM) and compared by unpaired *t*-test, linear regression *t*-test, Welch's *t*-test, Pearson's chi-square test, Fisher's exact test, Wilcoxon test, ordinary one-way ANOVA or two-way ANOVOA. $p < 0.05$ was considered statistically significant. ns, not significant; *$p < 0.05$, **$p < 0.01$, ***$p < 0.001$, and ****$p < 0.0001$.

## Acknowledgements

We would like to thank OrigiMed for the kind help of NGS and Nanjing University Chao Yan laboratory for the generous donation of KATOIII cell line.This work was funded by Grants from National Natural Science Foundation of China (82403862, 82403835), Jiangsu Provincial Natural Science Foundation Youth Project (BK20230151, BK20240247), Jiangsu Provincial Major Science and Technology Program (BG2024026), the Wu Jieping Medical Foundation Special Fund for Targeted Cancer Therapy (Youth Research Project) (320.6750.2023-11-30), Jiangsu Provincial Youth Science and Technology Talent Support Project (JSTJ-2025-715) and General Project of Nanjing Health Science and Technology Development Program (YKK24084).

# Additional information

## Funding

| Funder | Grant reference number | Author |
|---|---|---|
| National Natural Science Foundation of China | 82403862 | Yue Wang |
| National Natural Science Foundation of China | 82403835 | Tao Shi |
| Jiangsu Provincial Natural Science Foundation Youth Project | BK20230151 | Yue Wang |
| Jiangsu Provincial Natural Science Foundation Youth Project | BK20240247 | Tao Shi |
| Jiangsu Provincial Major Science and Technology Program | BG2024026 | Yue Wang |
| the Wu Jieping Medical Foundation Special Fund for Targeted Cancer Therapy (Youth Research Project) | 320.6750.2023-11-30 | Yue Wang |
| Jiangsu Provincial Youth Science and Technology Talent Support Project | JSTJ-2025-715 | Yue Wang |
| General Project of Nanjing Health Science and Technology Development Program | YKK24084 | Tao Shi |

The funders had no role in study design, data collection and interpretation, or the decision to submit the work for publication.

## Author contributions

Yue Zhang, Data curation, Formal analysis, Validation, Investigation, Writing – original draft; Hanbing Wang, Yunfeng Pan, Xueru Song, Data curation; Yutao Wei, Visualization; Jie Shao, Lixia Yu, Tao Shi, Supervision; Yue Wang, Resources, Supervision, Funding acquisition, Writing – review and editing

## Author ORCIDs

Yue Zhang (iD) https://orcid.org/0009-0005-3041-6300
Yue Wang (iD) https://orcid.org/0000-0002-3073-2574

## Ethics

Human sample used in this study obtained the patients' consent and the approval of the Ethics Committee of Nanjing Drum Tower Hospital (No. 2021-324-01).
All animal experiments were approved by the Institutional Animal Care and Use Committee of Drum Tower Hospital (approval number: 2022AE01029).

Reviewer #1 (Public review): https://doi.org/10.7554/eLife.104060.3.sa1
Reviewer #2 (Public review): https://doi.org/10.7554/eLife.104060.3.sa2
Reviewer #3 (Public review): https://doi.org/10.7554/eLife.104060.3.sa3
Author response https://doi.org/10.7554/eLife.104060.3.sa4

# Additional files

## Supplementary files

Supplementary file 1. Clinical characteristics of GC patients in Nanjing Drum Tower Hospital cohort

with and without *FGFR2* amplification.

MDAR checklist

## Data availability

Patients' data were from The Cancer Genome Atlas Program (TCGA), https://doi.org/10.1038/nature13480. All data generated or analyzed during this study are included in the manuscript and supporting files; source data files have been provided for all figures.

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
