## [Editor Report · eLife Assessment]

Based on the perceived low efficacy of current therapies targeted to FGFR2 in gastric cancer (GC), the authors investigate an approach that combines SHP2 inhibition with existing FGFR2 inhibitors. The data were largely collected and analysed using **solid** and validated methodology. There is some **useful** data regarding combination therapy in a new clinical cohort, which supports previous studies that have reported the potential of targeting RTKs together with phosphatases.

---

## [Referee Report · Reviewer #1 (Public review)]

The manuscript entitled "Blocking SHP2 1 benefits FGFR2 inhibitor and overcomes its resistance in 2 FGFR2-amplified gastric cancer" by Zhang, et al., reports that FGFR2 was amplification in 6.2% (10/161) of gastric cancer samples and that dual blocking SHP2 and FGFR2 enhanced the effects of FGFR2 inhibitor (FGFR2i) in FGFR2-amplified GC both in vitro and in vivo via suppressing RAS/ERK and PI3K/AKT pathways. Furthermore, the authors also showed that SHP2 blockade suppressed PD-1 expression and promoted IFN-γ secretion of CD8+ 46 T cells, enhancing the cytotoxic functions of T cells. Thus, the authors concluded that dual blocking SHP2 and FGFR2 is a compelling strategy for treatment of FGFR2-amplified gastric cancer. Although the finding is interesting, the finding that FGFR2 is amplified in gastric cancer and that FGFR inhibitors have some effect on treating gastric cancer is not novel. The data quality is not high, the effects of double inhibitions are not significant. It appears that the conclusions are largely overstated, the supporting data is weak and not compelling.

The data in Figure 1 is not novel; similar data have been reported elsewhere.

It is unclear why the two panels in fig 2a and 2b can not be integrated into one panel, which will make it easier to compare the activities.

The synergetic effects of azd4547 and shp099 are not significant in Fig 2e and 2f, as well as in Fig. 3g and Fig. 4f

Data in Fig. 5 is weak and can be removed. It is unclear why FGFR inhibitor has some activities toward t cells since t cells do not express FGFR.

---

## [Referee Report · Reviewer #2 (Public review)]

Summary:

This manuscript reports the application of a combined targeted therapeutic approach to gastric cancer treatment. The RTK, FGFR2 and the phosphatase, SHP2 are targeted with existing drugs; AZD457 and SHP099, respectively. Having shown increased mRNA levels of FGFR2 and SHP2 in a patient population and highlighted the issue of resistance to single therapies the combination of inhibitors is shown to reduce cancer-related signalling in two gastric cell lines. The efficacy of the dual therapy is further demonstrated in a single patient case study and mouse xenograft models. Finally, the rationale for SHP2 inhibition is shown to be linked to immune response.

Strengths:

The data is generally well presented, and the study invokes a novel patient data set which could have wider value. The study provides additional evidence to support the combined therapeutic approach of RTK and phosphatase inhibition.

Weaknesses:

Combined therapy approaches targeting RTKs and SHP2 have been widely reported. Indeed, SHP099 in combination with FGFR inhibitors has been shown to overcome adaptive resistance in FGFR-driven cancers. Furthermore, the inhibition of SHP2 has been documented to have important implications in both targeting proliferative signalling as well as immune response. Thus, it is difficult to see novelty or a significant scientific advance in this manuscript. Although the data is generally well presented, there is inconsistency in the interpretation of the experimental outcomes from ex vivo, patient and mouse systems investigated. In addition, the study provides only minor or circumstantial understanding of the dual mechanism.

Using data from a 161 patient cohort FGFR2 was identified as displaying amplification of FGFR2 in ~6% with concomitant elevation of mRNA of patients which correlated with PTPN11 (SHP2) mRNA expression. The broader context of this data is of value and could add a different patient demographic to other data on gastric cancer. However, there is no detail on patient stratification or prior therapeutic intervention.

Comments on revisions: This has been attended to in the revised version

In SNU16 and KATOIII cells the combined therapy is shown to be effective and appears to be correlated with increase apoptotic effects (i.e. not immune response).

Fig 2E suggests that the combined therapy in SNU16 cells is little better than FGFR2-directed AZD457 inhibitor alone, particularly at the higher dose.

The individual patient case study described via Fig 3 suggests efficacy of the combined therapy (at very high dosage), however the cell biopsies only show reduced phosphorylation of ERK, but not AKT. This is at odds with the ex vivo cell-based assays. Thus, it is not clear how relevant this study is.

The mouse xenograft study shows a convincing reduction in tumor mass/volume and a clear reduction in pAKT, whilst pERK remains largely unaffected by the combined therapeutic approach. This is in conflict with the previous data which seems to show the opposite effect.

Comments on revisions: The authors have clarified this point

In all, the impact of the dual therapy is unclear with respect to the two pathways mediated by ERK and AKT.

Finally, the authors demonstrate the impact of SHP2 on PD-1 expression and propose that the SHP099/AZD4547 combination therapy significantly induces the production of IFN-γ in CD8+ T cells. This part of the study is unconvincing and would benefit from an investigation of the tumor micro-environment to assess T cell infiltration.

---

## [Referee Report · Reviewer #3 (Public review)]

Summary:

Fibroblast growth factor receptor 2 (FGFR2) is a receptor tyrosine kinase that can be amplified in gastric cancer and serves as a potential therapeutic target for this patient population. However, targeting FGFR2 has shown limited efficacy. Thus, this study seeks to identify additional molecules that can be effectively targeted in FGFR2 amplified gastric cancer, with a focus on Src homology region 2-containing protein tyrosine phosphatase 2 (SHP2). The authors first demonstrate that 6% of gastric cancer patients in a cohort of human patient samples exhibit FGFR2 amplification. Furthermore, they demonstrate that FGFR2 mRNA expression is positively correlated with PTPN11 gene expression (which is the gene that encodes the SHP2 protein). Using human gastric cancer cell lines with amplified FGFR2, the authors then test the effects of combining the FGFR inhibitor AZD4547 with the SHP2 inhibitor SHP099 on tumor cell death and signaling molecules. They demonstrate that combining the two inhibitors is more effective at tumor cell killing and reducing activation of downstream signaling pathways than either inhibitor alone. In further studies, the authors obtained gastric cancer cells with FGFR2 amplification from a patient that was treated with FGFR2 inhibitor. While this patient initially showed a partial response, the patient ultimately progressed, demonstrating resistance to FGFR2 inhibition. Following isolation of tumor cells from the patient's ascites, the authors demonstrate that these cells are sensitive to the combination treatment of AZD4547 and SHP099. Further studies were performed using a xenograft model using athymic nude mice in which the combination of SHP099 and AZD4547 were found to reduce tumor growth more significantly than either treatment alone. Finally, the authors demonstrate using an in vitro culture model that this combination treatment enhances T cell mediated cytotoxicity. The authors conclude that targeting FGFR2 and SHP2 represents a potential combination strategy in gastric patients with FGFR2 amplification.

Strengths:

The authors demonstrate that FGFR2 amplification positively correlates with PTPN11 in human gastric cancer samples, providing a rationale for combination therapies. Furthermore, convincing data are provided demonstrating that targeting both FGFR and SHP2 is more effective than targeting either pathway alone using in vitro and in vivo models. The use of cells derived from a gastric cancer patient that progressed following treatment with an FGFR inhibitor is also a strength. The findings from this study support the conclusion that SHP2 inhibitors enhance the efficacy of FGFR-targeted therapies in cancer patients. This study also suggests that targeting SHP2 may also be an effective strategy for targeting cancers that are resistant to FGFR-targeted therapies.

Weaknesses:

The main caveat with these studies is the lack of an immune competent model with which to test the finding that this combination therapy enhances T cell cytotoxicity in vivo.

---

## [Author Response]

The following is the authors’ response to the original reviews

**Public Reviews:**

**Reviewer #1 (Public review):**
The data in Figure 1 is not novel, similar data has been reported elsewhere.

We are grateful for the critical evaluation of our finding. Although there have been a few researches indicating the prevalence of FGFR2-amplified GC patients, our research provided a novel dataset of 161 GC patients using next-generation sequencing (NGS) in China, further emphasizing the high frequency of FGFR2 amplification in gastric cancer patients. Moreover, the proportion of FGFR2-amplified GC patients in our center (6.2%) is relatively higher than that of TCGA cohort (5%).

We have transferred the original Figure 1C and 1D to the supplementary figures, and constructed a novel pie chart for Nanjing Drum Tower Hospital cohort to compare with the TCGA cohort.

It is unclear why the two panels in Fig 2a and 2b can not be integrated into one panel, which will make it easier to compare the activities.

Thanks for pointing this out. In the first figure of Figure 2a and 2b, we performed gradient concentration CCK8 detection on the cytotoxicity of SHP099 against tumor cells. In the second figure, we selected 10 μm (IC50) as the fixed concentration of SHP099 for combined efficacy testing with gradient concentration of AZD4547. Moreover, the units of the horizontal axis in both figure 2a and 2b cannot be unified. Therefore, we believe that the two figures in figures 2a and 2b are not suitable for merging into one figure.

For the convenience of observation, we integrated the first panel of figure 2a and 2b into one panel, and integrated the second panel in the same way.

The synergetic effects of azd4547 and shp099 are not significant in Fig 2e and 2f, as well as in Fig. 3g and fig. 4f

In Fig 2e and 2f, we not only analyzed the synergetic effects of 3 nM (a relatively lower dose) AZD4547 and 10 μm SHP099, but also 10 nM (a relatively higher dose) AZD4547 and 10 μm SHP099. The synergetic effects of different dosage combinations should be compared correctly. From our perspective, the combination treatment led to a stronger inhibition of phospho-FGFR, phospho-SHP2 and FGFR2-initiated downstream signaling molecules, especially in KATOIII.

For ease of comparison, we circled 10 μm SHP099, 10nM AZD4547 and 10nM AZD4547+10 μm SHP099 in red.

**Author response image 1. sa4fig1:** 

**Author response image 2. sa4fig2:** 

We also circled 10μM SHP099, 3nM AZD4547 and 3nM AZD4547+10 μm SHP099 in blue.

**Author response image 3. sa4fig3:** 

**Author response image 4. sa4fig4:** 

For ease of comparison, we also conducted grayscale value analysis and normalization using image J.

**Author response image 5. sa4fig5:** 

**Author response image 6. sa4fig6:** 

**Author response image 7. sa4fig7:** 

**Author response image 8. sa4fig8:** 

In Fig. 3g, the combination therapy exhibited relatively stronger inhibitory effects on phospho-ERK, phospho-AKT and phospho-mTOR.

For ease of comparison, we conducted grayscale value analysis and normalization using image J.

The unclear effect of combination therapy may be due to the presence of impurities other than tumor cells in patient’s ascites.

**Author response image 9. sa4fig9:** 

In Fig. 4f, it was obvious that phospho-AKT and phospho-mTOR were further suppressed in combination group.

For ease of comparison, we conducted grayscale value analysis and normalization using image J.

**Author response image 10. sa4fig10:** 

Therefore, in our opinions, our data could relatively sufficiently confirm the synergetic effects of AZD4547 and SHP099.

Data in Fig. 5 is weak and can be removed. It is unclear why FGFR inhibitor has some activities toward t cells since t cells do not express FGFR.

The activation effect of SHP099 on T cells has been validated in many articles. In a previous study published in Cancer Immunology Research, it was pointed out that the combination of FGFR2 inhibitor erdafitinib and PD-1 antibody can activate T cells and downregulate T cell surface exhaustion related factors (including PD-1) in vivo Therefore, the anti-tumor immune effect of FGFR2 inhibitor cannot be ignored. Although T cells do not express FGFR, FGFR2 inhibitors may still affect PD-1 expression on the surface of T cells in some other ways, which requires further research. We have deleted fig.5D in our article. We believe that the combination of FGFR2 inhibitor and SHP2 inhibitor not only has a direct killing effect on tumor cells, but also promotes anti-tumor immunity by activating T cells. Therefore, we believe that the in vitro data in Figure 5 is also meaningful.

**Reviewer #2 (Public review):**
Strengths:The data is generally well presented and the study invokes a novel patient data set which could have wider value. The study provides additional evidence to support the combined therapeutic approach of RTK and phosphatase inhibition.

We sincerely thank the reviewer for the critical evaluation and appreciation of our findings.

Weaknesses:Combined therapy approaches targeting RTKs and SHP2 have been widely reported. Indeed, SHP099 in combination with FGFR inhibitors has been shown to overcome adaptive resistance in FGFR-driven cancers. Furthermore, the inhibition of SHP2 has been documented to have important implications in both targeting proliferative signalling as well as immune response. Thus, it is difficult to see novelty or a significant scientific advance in this manuscript. Although the data is generally well presented, there is inconsistency in the interpretation of the experimental outcomes from ex vivo, patient and mouse systems investigated. In addition, the study provides only minor or circumstantial understanding of the dual mechanism.

We acknowledge that our research on the mechanism of dual inhibition is not deep enough. There remain more in-depth mechanisms of the combination of SHP2 inhibitor and RTK inhibitors needed to be explored, and it would be the main direction of our future study.

Using data from a 161 patient cohort FGFR2 was identified as displaying amplification of FGFR2 in ~6% with concomitant elevation of mRNA of patients which correlated with PTPN11 (SHP2) mRNA expression. The broader context of this data is of value and could add a different patient demographic to other data on gastric cancer. However, there is no detail on patient stratification or prior therapeutic intervention.

Thanks for pointing this out and we have added a table on patients’ stratification such as age, gender and so on. Unfortunately, data on patients’ prior therapeutic intervention weren’t collected.

In SNU16 and KATOIII cells the combined therapy is shown to be effective and appears to be correlated with increased apoptotic effects (i.e. not immune response).Fig 2E suggests that the combined therapy in SNU16 cells is a little better than FGFR2-directed AZD457 inhibitor alone, particularly at the higher dose.The individual patient case study described via Fig 3 suggests efficacy of the combined therapy (at very high dosage), however, the cell biopsies only show reduced phosphorylation of ERK, but not AKT. This is at odds with the ex vivo cell-based assays. Thus, it is not clear how relevant this study is.The mouse xenograft study shows a convincing reduction in tumor mass/volume and clear reduction in pAKT, whilst pERK remains largely unaffected by the combined therapeutic approach. This is in conflict with the previous data which seems to show the opposite effect. In all, the impact of the dual therapy is unclear with respect to the two pathways mediated by ERK and AKT.

Thank you for the comment. Previous researches have confirmed that both RAS/ERK and PI3K/AKT pathways are two important downstream signaling of FGFR2. In Fig 2E and F, we observed that in FGFR2-amplified cell lines dual blockade had significant inhibitory effects both on p-ERK and p-AKT, and the inhibitory effect on p-ERK is greater than that on p-AKT. Similarly, in Fig 3G, dual blockade mainly suppressed p-ERK, and slightly inhibited p-AKT and p-mTOR in cancer cells derived from the individual patient. Thus, in the two types in-vitro models, dual inhibition simultaneously inhibited both RAS/ERK and PI3K/AKT pathways, and primarily inhibited RAS/ERK pathway, which is not contradictory.

**Author response image 11. sa4fig11:** 

**Author response image 12. sa4fig12:** 

**Author response image 13. sa4fig13:** 

For the in-vivo animal model. Although dual inhibition had inhibitory effects on both pathways, it mainly suppressed p-AKT.

In both in vivo and in vitro models, combination therapy has a certain inhibitory effect on the RAS/ERK and PI3K/AKT pathways, but the emphasis on the two is not the same in vivo and in vitro. Considering the significant differences between in vivo and in vitro models, we believe that this difference in emphasis is understandable.

**Author response image 14. sa4fig14:** 

Finally, the authors demonstrate the impact of SHP2 on PD-1 expression and propose that the SHP099/AZD4547 combination therapy significantly induces the production of IFN-γ in CD8+ T cells. This part of the study is unconvincing and would benefit from the investigation of the tumor micro-environment to assess T cell infiltration.

To investigate the tumor micro-environment to assess T cell infiltration, we have to establish our research model in immunocompetent mice. However, there is currently only one type of gastric cancer cell line derived from mice, MFC, which is not a cell line with FGFR2 amplification. We attempted to transfect FGFR2 amplification plasmids into MFC, but the transfection effect was poor, making it difficult to conduct in vivo animal experiments.

**Reviewer #3 (Public review):**
Strengths:The authors demonstrate that FGFR2 amplification positively correlates with PTPN11 in human gastric cancer samples, providing rationale for combination therapies. Furthermore, convincing data are provided demonstrating that targeting both FGFR and SHP2 is more effective than targeting either pathway alone using in vitro and in vivo models. The use of cells derived from a gastric cancer patient that progressed following treatment with an FGFR inhibitor is also a strength. The findings from this study support the conclusion that SHP2 inhibitors enhance the efficacy of FGFR-targeted therapies in cancer patients. This study also suggests that targeting SHP2 may also be an effective strategy for targeting cancers that are resistant to FGFR-targeted therapies.Weaknesses:The main caveat with these studies is the lack of an immune competent model with which to test the finding that this combination therapy enhances T cell cytotoxicity in vivo. Discussing this limitation within the context of these findings and future directions for this work, particularly since the combination therapy appears to work quite well without the presence of T cells in the environment, would be beneficial.

Thank you for the great suggestion. To investigate the tumor micro-environment to assess T cell infiltration, we have to establish our research model in immunocompetent mice. However, there is currently only one type of gastric cancer cell line derived from mice, MFC, which is not a cell line with FGFR2 amplification. We attempted to transfect FGFR2 amplification plasmids into MFC, but the transfection effect was poor, making it difficult to conduct in vivo animal experiments.

**Recommendations for the authors:**

**Reviewer #1 (Recommendations for the authors):**
Minor points. The manuscript is poorly written and loaded with language errors.

We sincerely thank you for your constructive suggestion and we are sorry for the mistake. We have polished the article and corrected these language errors.

**Reviewer #2 (Recommendations for the authors):**
In addition to the comments made in the Public Review the manuscript lacks detail on statistical analysis of experimental results.

Thank you for your advice. In response to the feedback, we have supplemented detail on statistical analysis of experimental results in the “Methods” part.

**Reviewer #3 (Recommendations for the authors):**
There are numerous grammatical errors throughout, and incorrect wording is used in some places (such as "syngeneic mouse tumor model" rather than "xenograft tumor model", line 253). Careful proofreading and editing of this manuscript is recommended.

Thank you for your suggestion. We have made corrections to the relevant content of the article.

AZD4547 is an FGFR-selective inhibitor and is not specific for FGFR2 as it also targets FGFR1 and FGFR3, this should be clarified in the text.

Thank you for rasing this point. We have clarified that AZD4547 is an FGFR-selective inhibitor targeting FGFR1-3 in the “Introduction” part.

The specific FGFR inhibitor(s) used to treat the patient with FGFR2 amplification, are the authors able to provide this information?

Thank you for raising this important issue. Indeed, due to the difficulty of small molecule drug development, the fastest clinical progress currently is in FGFR pan inhibitors. Recently, Relay Therapeutics has also developed a highly FGFR2-selective inhibitor, RLY-4008, in phase I/II clinical trials, but lacks preclinical research on gastric cancer.

Figure 2F: the p38 and p-p38 bands are cut off at the bottom

We sincerely thank you for your thoughtful feedback. we have improved our experimental methods and retested the two p38 and p-p38 in Figure 2F by western blotting.

**Author response image 15. sa4fig15:**